# A realist review of health passports for Autistic adults

**Rebecca Ellis[1], Kathryn Williams[2,3], Amy Brown[1], Eleanor Healer[4], Aimee Grant[1]***

**1** Centre for Lactation Infant Feeding and Translation (LIFT), Swansea University, Swansea, United Kingdom, **2** Autistic UK CIC, Llandudno, Wales, United Kingdom, **3** School of Social Sciences, Cardiff University, Cardiff, United Kingdom, **4** School of Health and Social Care, Swansea University, Swansea, United Kingdom

\* aimee.grant@swansea.ac.uk

**Data Availability Statement:** All relevant data are within the paper and its Supporting information files.

**Funding:** AG, KW, AB and EH received funding for this research from the Swansea University Accelerate Health Tech Centre. Reference: 07/09/

## Abstract

### Background

Autism is a normal part of cognitive diversity, resulting in communication and sensory processing differences, which can become disabling in a neurotypical world. Autistic people have an increased likelihood of physical and mental co-occurring conditions and die earlier than neurotypical peers. Inaccessible healthcare may contribute to this. Autism Health Passports (AHPs) are paper-based or digital tools which can be used to describe healthcare accessibility needs; they are recommended in UK clinical guidance. However, questions remained as to the theoretical underpinnings and effectiveness of AHPs.

### Methods

We undertook a systematic literature search identifying studies focused on AHPs for adults (aged over 16 years) from five databases. Included literature was subjected to realist evaluation. Data were extracted using a standardised form, developed by the research team, which considered research design, study quality for realist review and the **C**ontext, **M**echanisms and **O**utcomes (**CMOs**) associated with each AHP tool.

### Findings

162 unique records were identified, and 13 items were included in the review. Only one item was considered high quality. Contextual factors focused on the inaccessibility of healthcare to Autistic patients and staff lack of confidence and training in supporting Autistic needs. **I**nterventions were heterogeneous, with most sources reporting few details as to how they had been developed. The most frequently included contents were communication preferences. **M**echanisms were often not stated or were inferred by the reviewers and lacked specificity. **O**utcomes were included in four studies and were primarily focused on AHP uptake, rather than **O**utcomes which measured impact.

### Conclusion

There is insufficient evidence to conclude that AHPs reduce the health inequalities experienced by Autistic people. Using an AHP tool alone in a healthcare **C**ontext that does not meet Autistic needs, without the inclusion of the local Autistic community developing the

21. RE's time was funded by this grant. The funders had no role in study design, data collection and analysis, decision to publish, or preparation of the manuscript.

**Competing interests:** The authors have declared that no competing interests exist.

tool, and a wider intervention to reduce known barriers to health inequality, may mean that AHPs do not trigger any **M**echanisms, and thus cannot affect **O**utcomes.

## Background

Autism has historically, and incorrectly, been pathologised in research and discussed using deficit-based language in line with the medical model [1]. Autism is a normal part of cognitive diversity, resulting in differences to communication style and sensory processing which are often experienced as disabling impairments in a neurotypical world [1], as described within a neurodiversity paradigm [2, 3]. Autistic people have worse health compared to their neurotypical peers [4]. This includes significantly worse physical and mental health and a lower life expectancy [5]. This is associated with systematic stigmatisation [6]—including by healthcare professionals [7]—and healthcare inaccessibility for Autistic people [8]. Autistic adults' experiences of health and healthcare have been severely under-explored in research [9].

Health passports (HP), also known as hospital passports, are a digital or physical source of information regarding patients' care needs and preferences designed to aid information transfer between patients and healthcare staff [10]. It has been argued that the use of HPs can address the differences in communication that may cause misunderstandings between patients and healthcare professionals, improve patient safety, and promote person-centred care [11]. HPs are many and heterogenous, and can include: personal details, contact information, communication needs, care needs, preferences, [12] signs of pain and distress, a medical history, and information on consent and capacity [11]. There is no current standardised approach to the development or implementation of HPs [11, 13–15] and a realist review of their use for medically complex children has found that more attention needs to be paid to the needs of patients, staff and organisations which are intending to use HPs [16].

Within healthcare systems, patient-centred tools have previously been developed and implemented to reduce barriers to care. For example, Greenberg et al., [17] evaluated the impact of the Asthma Passport, a patient-centred tool, finding it both significantly increased the number of patients who completed an Asthma Action Plan (AAP) and Asthma Control Test (ACT) within a clinical setting, and increased satisfaction with the care itself. Similarly, "Kardio-Passes", a patient passport containing relevant diagnostic and treatment data for those patients who have experienced cardiovascular diseases (CVD), have been rated by patients as a helpful tool for documenting follow-up data in rehabilitation [18]. It may be that the potential for observable clinical deterioration, or even death, related to uncontrolled asthma and cardiovascular diseases have been responsible for these passport type tools leading to benefits in these clinical specialities and in primary care clinics. Another example of patient-centric tools are various birth plan templates which aim to increase child-birth related patient satisfaction. Collaborative birth plans have been associated with positive birth **O**utcomes, [19] however other research has shown a reduction in patient satisfaction and control [20] and insufficient evidence to support or refute that birth plans improve birth experiences or increase satisfaction [21].

Autism-specific HPs (AHPs) have been recommended within UK policy documents [22–24] and NICE clinical guidelines [25]. AHPs are recommended to support Autistic people to access more equitable healthcare. Some policies suggest the use of generic HP tools, for example, the Welsh Government Autism Code of Practice, [24] encourages NHS Health Boards and Trusts to use the *Once for Wales Health Passport*. In their guidance notes for health professionals, [26] it is recommended that staff familiarise themselves with the contents of the profile and inform their colleagues of important care-related details, as well as noting that any changes should be made by staff on the HP. Examples of specific AHPs for adults within Europe

include tools developed by the UK National Autistic Society [27] and the *Healthcare Passport for Children and Adults on the Autism Spectrum* developed by Ireland's National Autism charity AsIAm [28]. Title two of The Americans with Disabilities Act prohibits discrimination against disabled people in both public and private hospitals and health facilities [29] and requires modified policies and procedures, such as the use of auxiliary aids, to improve communication and remove barriers to accessing care. In North America, the *Autism Healthcare Accommodations Tool (AHAT)* was developed as part of *the AASPIRE Healthcare toolkit* [30] and has been referenced within the Australian Journal of General Practice, [31] alongside the NAS *My Health Passport* as examples of resources to aid in the provision of care for Autistic people. In addition, the *My Autism Passport App* has been introduced to Canada [32].

The potential of AHPs to reduce health inequalities is often stated without reference to evidence of their effectiveness in policy documents, [22] clinical guidance, [25] and guidelines accompanying AHPs within individual health services [33]. When the use of AHPs for adults is evaluated, evidence of limited usage and implementation challenges can be found, [34–36] including one survey which identified only 4% of respondents used an AHP, but 30% would use one if given by their primary care doctor [37]. The theoretical underpinnings of AHP interventions have not yet been thoroughly considered. Throughout the rest of this article, we refer to all health passports as HPs, even if they are Autism specific, because many have multiple intended populations.

## The case for a realist evaluation

Realist evaluation has been used to understand what happens when interventions, especially those relating to health, are applied within particular contexts [38]. **C**ontext is defined as the observable social, economic, political, and cultural structures which in turn inform if **M**echanisms are triggered or not [39]. This is in recognition of the fact that interventions can flourish in one context, but fail to produce their intended outcomes in another, such as in relation to breastfeeding peer support which has not been successful in a UK experimental context [40]. A realist evaluation is a framework through which researchers systematically track and recognise the context in which the intervention is being delivered, how the intervention is delivered, and if the associated theoretical "mechanisms" underlying theories of change are achieved. **M**echanisms describe how the resources available influence the behaviour and thinking of those around an intervention [41]. Together, this evaluation provides valuable information alongside outcome data [42]. **O**utcomes refer to the intended, unintended, or unanticipated end results of the intervention being studied [41]. The focus is on quantitative **O**utcomes, although due to a dearth of quantitative data, we also present qualitative **O**utcomes in this research. Based on our knowledge of the evidence around HPs prior to this review, our initial programme theory was that the **C**ontext around HPs, including system-level challenges such as understaffing and inadequate appointment times, alongside low staff knowledge and confidence in supporting Autistic patients was not going to be easy or simple to improve. Therefore, interventions would need to be based on clear theories of change and explicit **M**echanisms of action to overcome the challenging **C**ontext. If these were not present, or the **M**echanisms were unable to fire, there would be little to no change in **O**utcome for Autistic people or healthcare staff as a result of using HPs. Based on this initial programme theory, coupled with poorer health **O**utcomes for Autistic adults compared to neurotypical peers, the scope and focus of the evaluation was narrowed to focus on HPs for Autistic adults only. To ensure all relevant evidence was included in our realist review, we decided to adopt a systematic literature search.

## Methodology

A systematic literature search and realist review was undertaken following guidance within the Preferred Reporting Guidelines for Systematic Reviews and Meta-Analyses (PRISMA), [43] and the RAMESES II reporting standards for realist reviews [44]. The protocol was prospectively registered with PROSPERO (registration ID: CRD42022304756).

### Aim

To review the evidence on HPs including theorising how they work, for whom and in what Contexts using realist evaluation methods [45].

### Community involvement, reflexivity and ethics

Three of the research team (RE, KW and AG) are Autistic Autism researchers. KW is also a director of Autistic UK, an organisation led by Autistic people for Autistic people. Additional input from the community was not sought prior to undertaking the review. Participant validation was not completed because of the complexity of the subject. Instead, we surveyed Autistic people about their views and experiences of HPs, which will be reported separately.[Grant, Unpublished] The remaining two members of the research team are a midwife (EH) and a professor of public health (AB). As the study relied entirely on published peer reviewed and grey literature, ethical approval was not sought.

### Search

We identified search terms by undertaking test searches based on hand searching keywords in relevant papers. The search strategy, developed with the support of a specialist librarian, involved two main terms, relating to (i) Autistic people and (ii) HPs (see Table 1). Five electronic databases were searched to ensure wide coverage within bio-medical and social science disciplines (Medline via OVID, PsychINFO via Ebscohost, CINAHL via Ebscohost, Web of Science via Clarivate, and SCOPUS via SciVal). Databases were searched from 2010 to January 2022 and limited to humans. Review articles identified in database searching were unpicked to identify additional papers.

**Study selection and eligibility criteria.** Papers were independently screened by two authors (RE and AG) at two stages: title and abstract; and full text. We did not exclude sources based on research design or methodology and commentaries were included. Due to limited resources for this study and the relative newness of AHPs, we included literature from the past 12 years. The following inclusion and exclusion criteria were utilised:

Inclusion criteria

**Table 1. Search terms.**

| Key/Mesh Term: | Alternative Terms: |
|---|---|
| Autism* (Inclusive of IDD—Intellectual and Developmental Disabilities). | "Autis*" OR "ASD*" OR "ASC" OR "Asperger*" OR "neurodevelopmental disorder" OR "pervasive developmental disorder" OR "PDD*" OR "IDD*" OR "Intellectual and developmental*" |
| Healthcare Passport* | "HP*" OR "Hospital passport*" OR "Communication passport*" OR "Patient passport*" OR "Healthcare Passport*" OR "Autism Passport*" OR "Traffic Light Passport*" OR "Traffic-light Passport*" OR "Care Passport*" OR "Bundle Care" OR "Education Healthcare Plan*" OR "EHCP*" |

- Population: Autistic adults ($\geq$16 years) and those involved in supporting access to healthcare or providing healthcare to Autistic adults

- Context: any healthcare setting

- Phenomenon: HPs that were either (i) exclusively for the use of Autistic people, or (ii) were for a broader population of patients **and** reported on their use with Autistic patients.

  Exclusion criteria

- Did not focus on HPs for Autistic adults ($\geq$one paragraph of relevant text) e.g.:

  - No passport-type tool described

  - Passports focused exclusively on something other than health, such as social care or education

  - HPs for non-Autistic populations

  - HPs for children aged under 16 years

- Published pre-2010

- Full text not available

- Full text not available in English

**Quality assessment, data extraction and realist synthesis methods.** Included sources were extracted independently by two authors (RE and AG) for core information relating to research design, if applicable (See Table 2), quality for realist review (See Table 3), and for contents relating to **C**ontext, **I**ntervention, **M**echanisms and **O**utcomes (See Table 4). Realist principles [45] were used to consider the impacts of **C**ontext on intervention components and intervention **M**echanisms [42] in acknowledgement that interventions will not work in all **C**ontexts, and that the 'messy' **C**ontext of healthcare can be particularly challenging [46] and thus impact on **O**utcomes. A summary document of the realist synthesis was presented to all authors to discuss and refine concepts and to identify areas of salience between sources. This was completed on two occasions, until consensus was reached.

## Results

A total of 162 unique records were identified through database searching, and 30 papers were sought for full-text review (see Fig 1). Two sources were unable to be retrieved via inter-library loan; 12 sources met the eligibility criteria. Ten reviews were identified and unpicked, identifying one additional paper, leading to a total of 13 items included in the review. Whilst the authors prefer identity-first language, and reject the notion that Autism is a disorder, we have used language from the original text when discussing the literature below.

Most of the included sources originated from the USA (n = 5) and the UK (n = 4) (see Table 2). Some of the tools described were *only* for Autistic patients (n = 6); others were not Autism-specific but they reported on health passport use by Autistic people (n = 7), as well as people with Intellectual Disabilities (ID), or Intellectual and Developmental Disabilities (IDD), referred to as a Learning Disability within a UK **C**ontext. Content that was not relating to the use of health passports by Autistic people was excluded from our analysis. The most common target populations for the HPs included: Two of the tools were focused on the transition from paediatric to adult healthcare providers [47, 48]. Most interventions were focused on a general hospital setting (n = 4), whilst others were for specific environments such as emergency

**Table 2. Summary of included papers.**

| Author, (year) country, and aims | Population, research participants and setting | Article type / research design | Data Collection and Analysis | Key Results/recommendations |
|---|---|---|---|---|
| **Blair (2013** [53]<br><br>UK<br><br>**Aim:** "To explore key issues in providing healthcare for people with IDs, how to minimise clinical risk and ensure care is appropriate, timely and lawful." (p.62) | **Population**: People with ID.<br><br>**Setting**: St. George's Hospital London. | Commentary | **Collection**: n/a<br>**Analysis**: n/a | **Recommendations**: Core reasonable adjustments (e.g.: no fixed visiting times); use HP; assess consent capacity; employ general care improvements for people with ID (e.g.: remember everyone's life has worth) |
| **Brasher, Middour-Oxler, Chambers and Calamaro (2020)** [50]<br><br>USA<br><br>**Aim:** "To summarise successful management approaches for children with ASD in paediatric ED, to identify ways to improve adult ED for individuals with ASD." (p.386) | **Population**: Autistic children and adults.<br><br>**Setting**: Emergency department. | Literature review | **Collection**: n/a<br>**Analysis**: n/a | **Results**: Interventions, including HP, can reduce barriers within the Autistic person and a lack of ASD specific staff training, to improve quality of care and empower nurses beyond paediatric or Autistic settings. |
| **Erickson Warfield, Crossman, Neumeyer, O'Brien and Kuhlthau (2017)** [47]<br><br>USA<br><br>**Aim**: To rate existing healthcare transition tools to identify tools for use in primary care clinics, and to develop a set of transition principles. | **Population**: Youth with special healthcare needs including ASD.<br><br>**Research participants**: Four paediatric and family medicine providers from community health centres.<br><br>**Setting**: Transition from paediatric to adult healthcare. | **Design**: Expert review of pre-existing tools. | **Collection**: Online survey, structured telephone interviews and a group conference call. Participants (n = 4) rated usefulness (yes/ no) for each tool.<br><br>**Analysis**: Frequency of advisors who would consider using each tool in their practice. Thematic analysis of qualitative data. | **Results**: Top rated tool was the 'Medical Summary and Emergency Care Plan' (the HP). No tool was viewed as ready for immediate patient use.<br><br>**Recommendations**: Balance standardisation of tools with individual needs; use transition champions to motivate staff; generic (not-Autism specific) transition protocol. |
| **Haidrani (2017)** [54]<br><br>Unknown<br><br>**Aim**: Not stated. | **Population**: Parents, Autistic children, and young people.<br><br>**Setting**: Unspecified. | Review of a mobile phone app (Android only). | **Collection**: n/a<br>**Analysis**: n/a | **Results**: Rated 4 stars. It can improve information sharing. |
| **Harris, Gorman, Doshi, Swope and Page (2021)** [48]<br><br>USA<br><br>**Aim:** "To address resource, training, and implementation gaps in healthcare transition for youth aged 12–21 years old with ASD through the development, implementation, and evaluation of a transition care tool within a Patient-Centred Medical Home (PCMH) practice." (p.755) | **Population**: Autistic paediatrics patients transitioning into adult services (251 patients aged 12–21 with ASD).<br><br>**Participants**: 'Quality Improvement Transition team' (n = 16); parents of Autistic children (n = 13); SNPCP staff.<br><br>**Setting**: Two outpatient Special Needs Primary Care Practices (SNPCP). | **Design**: Quality improvement study. | **Collection**:<br><br>**Pre**: Survey of parents of Autistic children (n = 13). Individual and group Discussions with staff. Health information technology systems assessment.<br><br>**Post**: Routine data collection related to intervention (number of auto-prompts, checklists completed); interviews with seven SNPCP providers.<br><br>**Analysis**: Not stated. | **Results**: Transition checklist prompted 100% of the time (449 appointments), completed 44% of the time. Social work Transition template used 179 times for a total of 112 patients, with an "average" of 1.6 contacts.<br><br>**Recommendations**: Longer appointments times; increase physician knowledge/comfort in discussing transitions. |

(*Continued*)

**Table 2.** (Continued)

| Author, (year) country, and aims | Population, research participants and setting | Article type / research design | Data Collection and Analysis | Key Results/recommendations |
|---|---|---|---|---|
| **Heifetz and Lunsky (2018)** [49] <br><br> Canada <br><br> **Aim**: To evaluate communication tools to be used by people with IDD in psychiatric and general emergency departments in three different regions of Ontario. | **Population**: People with IDD and their families. <br><br> **Participants:** <br><br> Interviews: Hospital clinical staff, community health and IDD service providers, community-based healthcare case coordinators, and one parent. Questionnaires: Caregivers/parents and individuals with IDD. <br><br> **Setting**: Ontario hospital emergency departments. | **Design:** Communication tools evaluated, locally tailored and implemented. | **Collection**: 18 semi-structured telephone interviews. 28 questionnaires (open and closed questions) were completed via post and email. <br><br> **Analysis:** Interviews: Thematic Analysis. <br><br> Questionnaires: Not stated. | **Results**: The tool was described as helpful for professionals and families, facilitative of communication and good service, easy to use, helpful, potentially helpful in other Contexts and for other populations, not consistently used and un-necessary additional work, with more information needed within it. <br><br> Staff generally rated the tool more favourably than the families, who were less optimistic. |
| **Kelbrick, Radley, Shaherbano, Cook and Simmons. (2014)** [34] <br><br> UK <br><br> **Aim**: To identify psychiatric and physical ill health and to introduce physical health screening and management in an adult male low secure ASD unit. | **Population**: Male, Autistic individuals. <br><br> **Setting**: St Andrew's Healthcare, UK. | **Design**: Quality improvement study with pre-and post-audit. | **Collection**: Case note audit (electronic and paper) of all service users (n = 16 at pre and n = 18 at post audit). Measure of physical activity (International Physical Activity Questionnaire). <br><br> **Analysis**: "Appropriate statistical tests." (p.32) | **Results**: 14 of 16 (pre) and 14 of 18 (post) patients had a physical health assessment. Limited use of the HP. |
| **Lalive d'Epinay Raemy and Paignon (2019)** [55] <br><br> Switzerland <br><br> **Aim**: To describe an interventional project in a University Hospital to enhance care for patients with IDDs in an acute care setting in Western Switzerland. | **Population**: Patients with IDD, ASD and severe disabilities. <br><br> **Participants**: *The Disability Project* multi-disciplinary team: health professionals, social workers, families, architects, social care providers, researchers. <br><br> **Setting**: Hopitaux Universitaires de Genève (HUG). | **Design**: Quality Improvement Project using >60 working group sessions. | **Collection**: Working group sessions. <br><br> **Analysis**: Not stated. | **Recommendations**: Central phone number, staff members to champion; use of admission sheet (HP tool); standardisation of electronic patient records; provide information to patients (e.g.: internet/easy read resources); training for healthcare professionals, ID-specific out-patient clinic. |
| **Learning Disability Practice (2014)** [56] <br><br> UK <br><br> **Aim**: N.A | **Population**: Autistic People. <br><br> **Setting**: Hospital Environments. | News article | **Collection**: n/a <br> **Analysis**: n/a | **Recommendations**: HPs may improve outcomes. |
| **Nicolaidis, Raymaker, McDonald, Kapp, Weiner, Askkenazy, Gerrity, Kripke and Platt (2016)** [30] <br><br> USA <br><br> **Aim:** "To use Community-Based Participatory Research (CBPR) to develop and evaluate an online healthcare toolkit for Autistic adults and their PCPs." (p.1180) | **Population**: Autistic adults and PCPs. <br><br> **Participants**: 259 Autistic adults and 51 PCPs. <br><br> **Setting**: Online. | **Design**: Development, piloting, refining and evaluation (pre-/ post-comparison) of toolkit. | **Collection:** **Development**: Cognitive Interviewing with Autistic adults, supporters, and PCPs using online audio computer-assisted survey. 2-week retest reliability study. **Evaluation**: Mixed-methods, single-arm pre/post intervention comparison. <br><br> **Analysis**: Cronbach's alphas for scored scales. Paired t-tests for pre- and post-intervention outcomes. Thematic analysis of open-text data. | **Results**: Increased self-efficacy, reduced barriers to accessing healthcare and strong support from participants for the toolkit. Patients thought the toolkit could change PCP behaviour but were frustrated when PCPs did not engage with the report. |

*(Continued)*

**Table 2.** (Continued)

| Author, (year) country, and aims | Population, research participants and setting | Article type / research design | Data Collection and Analysis | Key Results/recommendations |
|---|---|---|---|---|
| **Perkins and Vanzant, (2019)** [52]<br><br>USA<br><br>**Aim:** "To highlight free health resources available from the Florida Center for Inclusive Communities (FCIC)." (p.49) | **Population**: People with IDD.<br><br>**Setting**: Online. | Commentary | **Collection**: n/a<br>**Analysis**: n/a | **Results**: Resources can improve communication, empower patients, and inform providers<br>**Recommendations**: Resources should be developed with stakeholders. |
| **Sajith, Teo, and Ling (2018)** [51]<br><br>Singapore<br><br>**Aim**: To develop and implement communication passports in an acute inpatient unit for adults with IDs. | **Population**: People with IDs and/ or Autism.<br><br>**Participants**: Patients, caregivers, staff from the ward.<br><br>**Setting**: Adult Neurodevelopmental Services (ANDS), Institute of Mental Health, a tertiary psychiatric hospital in Singapore. | **Design**: Development, piloting, and refinement of a communication passport. | **Collection**: **Development**: Focus groups with project team.<br>**Evaluation**: "feedback" from patients, carers, and staff.<br><br>**Analysis**: Not stated | **Results**: Communication Passport found to be useful by hospital and social care staff, but lack of training in the community reduced the utility of some strategies. |
| **Unitt, (2018)** [57]<br><br>UK<br><br>**Aim**: Not stated. | **Population**: People with LDs.<br><br>**Setting**: Hospital Environments. | Blog Post | **Collection**: n/a<br>**Analysis**: n/a | **Recommendations**: Misuse of passports is a safety concern, not fitting the "best interest" of individuals, under the Mental Capacity Act. |

Key: ASD: Autism Spectrum Disorder; ID: Intellectual Disability; ED: Emergency Department; LD: Learning Difficulties/Disability; IDD: Intellectual and Developmental Disability; PCP: Primary Care Providers

departments (EDs) (n = 2) [49, 50]. Most sources reported on a single HP tool (n = 8); others were part of a wider toolkit [50] or intervention [48]. Five sources discussed or evaluated tools which sought to improve communication [34, 48–51]. These included the testing and re-testing of a particular intervention [34] or piloting an intervention and seeking stakeholder feedback [34]. Four sources did not report a research study, and were descriptive in nature, including commentaries and blog posts. Other sources described research or quality improvement initiatives but did not include details regarding data collection and analysis [34] or described a tool without detailing its evaluation or creation [52].

## Quality assessment for realist review

Table 3 outlines the quality of the sources for realist review. Three papers included limited detail relating to theories of change, [49, 55, 51] with one providing sufficient detail [30]. Within the sources and their associated appendices, almost half described their HP in detail (n = 6), five to some extent, and two provided no details. Intervention components in addition to the HP tool included training and additional environmental accommodations, which were described in detail (n = 4), to some extent (n = 5) or not at all (n = 3). The quality of information on the social **C**ontext surrounding healthcare for Autistic people was high quality in five sources and not provided in another five sources; the other three papers were not centred on Autistic individuals. Six papers discussed issues surrounding implementation, with a further three having some information in this category. Two papers outlined a process evaluation for the HP itself [30] or the wider intervention [48]. Five papers were not empirical in nature and

**Table 3. Quality of studies for realist evaluation.**

| Materials related to HP | Intervention theory described in detail? | HP described in detail | Intervention components around HP described in detail? | Social Context relating to healthcare for Autistic people well described? | Implementation issues discussed? | Process evaluation? (No of staff trained, HPs given out, HPs used etc) | Measured Outcome? | Measured Outcome consistent with intervention theory? |
|---|---|---|---|---|---|---|---|---|
| Blair [53] | No | Some | Some | No | Some | No | No | n/a |
| Brasher [50] | No | Yes | Some | Yes | n/a | n/a | n/a | n/a |
| Erickson [47] | No | Some | n/a | Yes | Yes | n/a | n/a | n/a |
| Haidriani [54] | No | Some | Some | No | n/a | n/a | n/a | n/a |
| Harris [48] | No | Some | Yes | Yes | Yes | Yes[1] | Yes | n/a |
| Heifetz and Lunsky [49] | Some | Yes | Yes | n/a[2] | Yes | No | Yes[3] | Yes |
| Lalive [55] | Some | Yes | Yes | n/a[2] | Yes | Unsure[6] | No[5] | n/a |
| Learning Disability Practice [56] | No | Some | No | No | n/a | n/a | n/a | n/a |
| Kelbrick [34] | No | No | No | Yes | Some | No | No[4] | n/a |
| Nicolaidis [30] | Yes | Yes | Yes | Yes | Yes | Yes | Yes | Yes |
| Perkins [52] | No | Yes | Some | n/a[2] | No | No | No | n/a |
| Sajith [51] | Some | Yes | Some | No | Yes | No | No | n/a |
| Unitt [57] | No | No | No | No | Some | n/a | n/a | n/a |

[1] The process evaluation was focused on the wider quality improvement tool, but the use of HPs was not included within this evaluation.

[2] Population was individuals with IDD.

[3] Questionnaires with 28 people, including 3 patients who described usefulness. Unclear how many patients received a HP, so not a clear **O**utcome measure.

[4] Numbers of participants in the intervention were described in a pre/post audit, but **O**utcome measures did not related to HPs.

[5] Lack of epidemiological data was identified as an issue in this study. We inferred that future efforts will include an attempt to gather robust **O**utcome measures.

[6] Almost all individuals have completed the tool, but other process information is limited.

another five reported no process evaluation. For one paper, it was noted participants had completed the tool, but no further process evaluation information was provided [55]. Furthermore, only three sources [30, 48, 49] reported an **O**utcome measure for the HP specifically, with one detailing **O**utcomes for their wider intervention [34]. Due to the lack of intervention theory and/or **O**utcome measures, in the majority of cases, it was not possible to state whether the **O**utcome measured was consistent with intervention theory. The exceptions to this were Nicolaidis et al.[30] and Heifetz and Lunsky [49]. Only one source Nico 2016 met all eight quality criteria for realist review.

## Realist evaluation

Within this section, we present the contents of the included sources following a **C**ontext, **I**ntervention, **M**echanisms and **O**utcomes (**CMO**) format, which is summarised for each study in Table 4.

All sources included some contextual elements, but these varied in their depth and the population, for example considering the **C**ontext for patients with ID. Interventions ranged from a singular HP tool to multi-stage (development and evaluation) and multi-component Interventions. There were also variations in the contents of the HP tools. **M**echanisms were more often

**Table 4. Context, mechanisms and outcomes of included papers.**

| Author: | Context: | Intervention: | Mechanisms: | Outcomes: |
|---|---|---|---|---|
| Blair [53] | Population: ID<br><br>Setting: St. George's Hospital London, UK.<br><br>Staff: Hospital clinicians.<br><br>Known Issues:<br>Hospital environments as "frightening" (p.58) for population.<br><br>Population more likely to need hospital care.<br><br>Clinicians have difficulties assessing capacity. | Development: Developed "in partnership" (p.60) with people with ID, carers and community/hospital staff.<br><br>Wider Intervention Name: N/A<br><br>HP: Hospital passports, adapted from the one created by the then Gloucestershire NHS Primary Care Trust.<br><br>Other Element: Core reasonable adjustments (e.g.: double appointments, no visiting times for carers).<br><br>Training: N/A.<br><br>Intended Use: Completed by individuals and family, prior to care commencing.<br><br>For clinicians to aid in care. | Explicit:<br>Enhances knowledge of patient, improving safety.<br><br>Embedded (95% of population use HP).<br><br>Implicit:<br>Humanising patients with ID. | Quantitative: N/A<br><br>Qualitative: Examples from practice report (1 Autistic patient). |
| Brasher et al. [50] | Population: ASD<br><br>Setting: Paediatric ED: Atlanta, USA.<br><br>Staff: "Child life specialists, social workers, and staff from the departments of nursing, physical therapy, occupational therapy, and medicine." (p388)<br><br>Known Issues: Barriers caused by alternative communication, altered sensory perception, lack of Autism specific training. | Development: N/A<br><br>Wider Intervention Name: N/A<br><br>HP: My Health Passport (National Autism Society) (Discussed, not Implemented)<br><br>Other Elements: Coping plans, sensory carts, kits and boxes, environmental modifications, team approach, multi-disciplinary team training, caregiver involvement and SCRAMBLE.<br><br>Training: Multidisciplinary<br><br>team training discussed as resource for adult EDs.<br><br>Intended Use: To enhance care within an adult ED. | Explicit:<br>Enhances knowledge in a fast-paced clinical environment.<br><br>Allows Autistic person to "explain their unique needs" (p388), resulting in patient-centred care.<br><br>Implicit: "Empower nurses" (p.386)<br><br>Increase quality of care. | Not introduced into practice. |
| Erickson [47] | Population: ASD.<br><br>Setting: Primary care. Massachusetts, USA.<br><br>Staff: Primary care physicians (PCPs) and nurses.<br><br>Known Issues: Transition from paediatric to adult care presents challenges.<br><br>Shrinking numbers of PCPs. | Development: N/A<br><br>Wider Intervention Name: N/A<br><br>Not an intervention study; describing and testing available tools including:<br><br>HP: "Medical summary and emergency care plan." (p.135)<br><br>Other Element: 12 transition tools described and rated.<br><br>Training: N/A<br><br>Intended Use: To ease transition from paediatric to adult services. | Explicit: PCPs do not prioritise care for ASD youth.<br><br>Implicit:<br>HPs **aim to** reduce burden of transition on PCPs, increasing likelihood of ASD youth securing an adult PCP. | Quantitative: n/a<br><br>Qualitative: n/a<br><br>Need:<br>Balance between standardisation and individualisation of HPs.<br><br>Transition champion/team.<br><br>Transition protocol for complex youth. |

(Continued)

**Table 4.** (Continued)

| Author: | Context: | Intervention: | Mechanisms: | Outcomes: |
|---|---|---|---|---|
| Haidrani [54] | Population: Parents, children and young people. | Development: Not detailed. | Explicit: Reduces anxiety for families. | Quantitative: N/A |
| | Setting: N/A | Wider Intervention Name: N/A | Implicit: N/A | Qualitative: N/A |
| | Staff: N/A | About Me (Autism passport) app. Available on android only. | | |
| | Known Issues: N/A | HP: This is me profile | | |
| | | Other Element: | | |
| | | Information about assessments and contacts with different services | | |
| | | Training: N/A | | |
| | | Intended Use: | | |
| | | Learning disability nurses to share with patients. | | |
| | | Patients to share information with health professionals during emergencies. | | |
| Harris et al. [48] | Population: Transitioning Autistic adolescents | Development: Transition team (range of stakeholders including healthcare professionals) tasked with creation of the ASD SNPCP transition programme, including processes, implementation and evaluation of process. | Explicit: | Quantitative: |
| | Setting: Two sites within a hospital system—outpatient Special Needs Primary Care Practices | | Increased familiarity and comfort with transition tools to parents and clinicians. | 449 adolescents well visits; 100% auto prompt to discuss transition; transition checklist complete in 44% of appointments. |
| | (SNPCP) recognised as Family-Centred Medical Homes. 3,100 patients, 634 patients diagnosed as Autistic. | Wider Intervention Name: Transition project (for all patients with any special need). Involving experts and stakeholders. | Electronic prompts for clinicians during appointments (did not mention HPs). | 17 transition only visits conducted (of 251 eligible Autistic young people). |
| | Staff: SNPCP staff members: medical providers, social workers, support staff, five nurses, a medical assistant, and patient care coordinators. | HP: Adapted Health Passport. | Implicit: | Social work template for transition used by 112 patients (total of 179 times). |
| | | Other Element:Transition reference sheet (for patient age range specifically) | NB: No mention of HP on clinician facing tool, so unlikely to become embedded. | Qualitative: |
| | Known Issues: | Practice checklist (for practice manager only). | | Reasons for lack of physician engagement: lack of time (especially for complex medical needs); discomfort with topic and lack of familiarity with insurance eligibility rules, patients' younger age, clinician not the patients' regular provider of care. |
| | Adult providers unwilling and/or untrained to accept care for Autistic youth. | Transition template (health professional facing) —does not mention HP. | | |
| | High proportion of co-occurring conditions in Autistic youth, including mental health. | Training: 1 month prior to implementation: • Half-day workshop for families (over 30 parents). | | |
| | Inadequate transition support | • All staff trained in utilizing the transition resources; reinforced through daily meetings and huddles. 1–1 follow ups to reinforce training. | | Physician recommendation: transition should be social work led. |
| | | Clinicians can refer to social worker for a "transition visit". (p.759) | | |
| | | Intended Use of HP: Unclear. | | |

*(Continued)*

**Table 4.** (Continued)

| Author: | Context: | Intervention: | Mechanisms: | Outcomes: |
|---|---|---|---|---|
| Heifetz and Lunsky [49] | Population: IDD, families.<br><br>Setting: Ontario hospital emergency departments (three regions).<br><br>Staff: Hospital and community staff; implementation facilitator.<br><br>Known Issues:<br><br>IDD: more likely to visit emergency departments and be hospitalized.<br><br>Barrier to care: information gaps—including communication barriers.<br><br>ED: rapid paced environment; less time for staff to support flow of information | Development: HP template selected and adapted to each community by the community working group.<br><br>Wider Intervention Name: N/A<br><br>HP: Adapted HP—variation in passport tool in each of three regions.<br><br>Other Element: Passport developed by hospital and community stakeholders in each of the three regions separately during "exploration and engagement stage", followed by "installation stage". (p.25)<br><br>Local implementation facilitator/champion.<br><br>Training: Unclear, including; community "orientation sessions" (p.25) to facilitate HP completion; and Hospital/Community agency "refresher workshops." (p.29)<br><br>Intended Use: Unclear. | Explicit:<br><br>Poor communication causes stress.<br><br>HPs improve consistency and quality of patient care, patient comfort, and reduce both unnecessary and return visits. HP awareness will facilitate use.<br><br>Local development of tool increases uptake.<br><br>Simple, easy to use and coordinated approach will increase embeddedness with staff and patients and reduce repetition of information given by patients/families.<br><br>Facilitator with clinical background adds credibility and relevant knowledge to implementation.<br><br>Systems required to facilitate storage and retrieval of HPs.<br><br>Implicit:<br><br>HPs designed to be evolving documents.<br><br>Strong leadership increases embeddedness. | Quantitative:<br><br>Evaluation participants (n = 28). Majority did not have a chance to use their HP (82%).<br><br>Positive feedback includes user friendliness (82%); "easy" (p.26) to complete (79%) clear instructions (68%), provides background information (75%) and makes the patient more comfortable (80%).<br><br>Helped: caregiver feel more involved and respected (65%), hospital staff adapt their care approach (65%), improve communication with staff (80%), and make decisions with better information (80%).<br><br>Qualitative:<br><br>Families were less optimistic than staff about the usefulness of the tool. Regional differences identified.<br><br>Tool: may be helpful for hospital, GP, agency staff, and families; facilitates communication and improves service; is a good summary and easy to use; (tool guidance is helpful; useful in other Contexts, for other populations.)<br><br>Challenges/barriers feedback, tool: more time needed to engage; not consistently used/forgotten/lack of its awareness; tool is unnecessary added work/need for additional information. |
| Kelbrick [34] | Population: ASD<br><br>Setting: St Andrew's Healthcare, Northampton, UK. ASD low secure unit (20 bed, male).<br><br>Staff: unclear.<br><br>Known Issues: ASD: increased risk of physical health conditions (e.g.: obesity).<br><br>Patients face sensory challenges.<br><br>Clinicians' lack of awareness ASD and of co-occurring conditions. | Development: Not detailed.<br><br>Wider Intervention Name: Quality<br><br>Enhancement measures of a HP and physical health screening/management guides<br><br>NB: responding to several clinical guidelines/policy documents.<br><br>HP: HPs.<br><br>Other Element: Pre and post audit. Evidence-based screening and management for physical health conditions.<br><br>Training: "informal staff education" (p.35) after initial audit—unclear if HP related.<br><br>Intended Use:<br><br>improve communication and increase involvement of patients. | Explicit: None (HP not the focus of intervention).<br><br>Implicit: N/A | Quantitative: N/A<br><br>Qualitative: Use of HP "limited" (p.35) and "disappointing" (p.36), due to lack of staff and patient understanding of HPs. |

*(Continued)*

**Table 4.** (Continued)

| Author: | Context: | Intervention: | Mechanisms: | Outcomes: |
|---|---|---|---|---|
| Lalive d'Epinay Raemy and Paignon [55] | Population: IDD, ASD and people with severe disabilities.<br><br>Setting: Geneva University Hospital.<br><br>Staff: Healthcare professionals and hospital staff.<br><br>Known Issues:<br><br>Lack of awareness by Healthcare professionals of specific health issues surrounding those with IDD.<br><br>Lack of communication and information transmission between hospital staff and service users.<br><br>Lack of IDD training in healthcare professionals and hospital staff.<br><br>Inaccessibility of hospital services and buildings for those with IDD.<br><br>Lack of epidemiological data. | Development: Developed by Communication Working Group.<br><br>Wider Intervention Name: The Disability Project (2012–2017).<br><br>HP: Disability Admission sheet.<br><br>Other Element: Waiting period in safe, dedicated space, central phone number for admissions, full time ID physician and nurse position, full time case manager, dedicated web page, training for key staff (3 levels: 15 minutes; 2 hours; 5 days), environmental accessibility (e.g.: ramps).<br><br>Training: Non-specific training for staff, HCP and hospital staff<br><br>Intended Use: Unclear who completes/updates it. Stored in the electronic patient management system. | Explicit:<br><br>Humanizing: Raising awareness of specific ID needs; previously "invisible." (p.7)<br><br>Embeddedness with staff, through training and electronic patient's management system.<br><br>Establish trust between PWID and hospital.<br><br>Improve communication.<br><br>Relieves repetition of information by parents/carers.<br><br>Aids hospital staff access information quickly.<br><br>Improve care: reduce emergency admissions and readmissions.<br><br>Leadership provided by ID specialist physician and "nurse case manager." (p.8)<br><br>Improve accessibility of buildings.<br><br>Implicit: N/A | Quantitative: has been filled in by all supported residential accommodations and families, for almost every PWID in Geneva. 1,017 patients between 2016–2018.<br><br>Qualitative:<br><br>Systematically used in the hospital. "Significantly improved" (p.7) communication. Objections: positive discrimination; additional costs.<br>• Access ramps, handrails and doors added in the Psychiatric hospital wing for wheelchair access.<br>• Eight new dedicated parking spots.<br>• The revolving doors of the main entrance to the main hospital have been replaced to improve access.<br><br>Increased connections between hospital staff and external partners and volunteers. |
| Learning Disability Practice [56] | Population: Autism.<br><br>Setting: English hospitals.<br><br>Staff: Nurses and hospital staff.<br><br>Known Issues: N/A | Development: N/A<br><br>Wider Intervention Name: N/A<br><br>HP: Hospital Passport<br><br>Other Element: N/A<br><br>Training: N/A<br><br>Intended Use: "To assist" staff. (p.6) | Explicit: None.<br><br>Implicit:<br><br>Patients and staff will be able to access HPs.<br><br>HP contains information on distress and communication of pain—implies staff will change behaviour. | Quantitative: N/A<br><br>Qualitative: N/A |

*(Continued)*

**Table 4.** (Continued)

| Author: | Context: | Intervention: | Mechanisms: | Outcomes: |
|---|---|---|---|---|
| Nicolaidis et al. [30] | Population: Autistic adults and PCPs.<br><br>Setting: Healthcare<br><br>Staff: "Healthcare providers"<br><br>Known Issues:<br><br>Scarce services and resources for Autistic adults.<br><br>Additional support required for co-occurring conditions and have a greater number of unmet health needs.<br><br>Autistic adults face multiple healthcare barriers, leading to lower use of preventative service and higher use of emergency services.<br><br>Primary care providers (PCPs) lack the training necessary to care for Autistic adults, leading to incorrect assumptions, an unwillingness to accommodate written communication and the use of inaccessible language. | Development: CBPR—and primary research including; a survey and qualitative research with Autistic people and PCPs.<br><br>Wider Intervention Name: The Academic Autism Spectrum Partnership in Research and Education (AASPIRE) Healthcare Toolkit.<br><br>HP: Autism Healthcare Accommodations Tool (AHAT)<br><br>Other Element: ". . .general healthcare and Autism-related information, checklists, worksheets, and other resources." (p.1180)<br><br>Modified version of tool is available for (i) Autistic people and (ii) supporters<br><br>Training: Online Q&A for Autistic people/ supporters. Additional training planned in the future.<br><br>Intended Use: Autistic patients, supporters and their PCPs. | Explicit:<br>"Potential leverage points" (p.1181) to be targeted identified from their research.<br><br>"Ensure relevance, utility and accessibility" (p.1181) through CBPR approach to development.<br><br>Personalised information on patients acknowledges heterogeneity of Autistic people.<br><br>Patient or carer completes a customised cover letter and report for PCP<br><br>Accommodations targeted throughout the healthcare journey, to remove as many barriers as possible, including making appointments, environment, communication, bodily awareness and incorporating supporters.<br><br>Increase self-efficacy in Autistic people<br><br>AHAT as a step on the journey to create systematic change to improve healthcare accessibility.<br><br>Implicit: N/A | Quantitative:<br>95% (of n = 126) said the toolkit was easy to understand. 92% (of n = 126) of Autistic people would recommend the toolkit to a friend; 95% would recommend to a health professional.<br><br>95% (of n = 126) of Autistic people said the toolkit was useful.<br><br>65% gave permission to mail completed AHAT to PCP.<br><br>43 participants saw PCP within 1 month; satisfaction increased (30.9 to 32.6, p = 0.03).<br><br>In pre/post-intervention comparisons, the total number of barriers encountered by patients decreased significantly (from a mean of 4.07 at baseline to 2.82 post-intervention; p <0.001).<br><br>Participants' self-efficacy in navigating the healthcare system also increased (37.92 to 39.39, p = 0.02).82% of (n = 39) PCPs rated toolkit as moderately or very useful. 87% (of n = 37) would recommend it to their patients.<br><br>Qualitative:<br><br>Toolkit described as:<br>• A means to clarify and communicate needs.<br>• Validating experiences.<br>• Empowering self-advocacy.<br>• Helping them prepare for visits.<br>• Suggesting new things to try.<br><br>Most participants enthusiastic that PCP behaviour may be changed. A minority were concerned that their PCP would have a negative response or would not use the AHAT report.<br><br>Many reported enthusiasm for and positive changes in providers or their staff.<br><br>Several PCPs noted that they already were doing what was recommended and two PCPs felt that they did not have time to implement accommodations. |

*(Continued)*

**Table 4.** (Continued)

| Author: | Context: | Intervention: | Mechanisms: | Outcomes: |
|---|---|---|---|---|
| Perkins and Vanzant [52] | Population: IDD<br><br>Setting: available online to be used widely.<br><br>Staff: Healthcare providers.<br><br>Known Issues:<br><br>Co-occuring conditions more likely.<br><br>Inadequate training around IDD for healthcare professionals.<br><br>Poor care leads to dissatisfaction amongst service users.<br><br>A need for nationwide systematic change and mandatory IDD training. | Development: N/A<br><br>Wider Intervention Name: N/A<br><br>HP: My Health Passport and My Health Report.<br><br>Other Element: N/A<br><br>Training: N/A<br><br>Intended Use: Health Passport is completed by the patient or caregiver and shared with their health provider. | Explicit:<br><br>To make "the unfamiliar familiar" (p.50)—unclear if this is for patients or staff.<br><br>Training reduces staff awkwardness.<br><br>Improved patient care through increased knowledge of disability and individual patients, including enhanced communication.<br><br>Skills and competencies need to be developed, agreed, and endorsed.<br><br>Implicit:<br><br>Endorsements by professional bodies will increase acceptability. | Quantitative: N/A<br><br>Qualitative: N/A |
| Sajith, Teo and Ling [51] | Population: Adults with moderate to severe ID and/or Autism with MH difficulties, particularly "challenging behaviours" (p.166) like aggressive and self-harming behaviours.<br><br>Setting: Adult Neurodevelopmental Services (ANDS) at the Institute of Mental Health, Singapore. The only psychiatric tertiary hospital in Singapore.<br><br>Staff: ANDS ward staff.<br><br>Known Issues:<br><br>Lack of effective communication system for those with severe IDs.<br><br>Communication style of those with complex needs may not be understood.<br><br>Staff often lack information about patient's care needs and (particularly non verbal) communication styles.<br><br>Staff struggle to debrief to parents and carers on discharge; high rates of staff turnover in community residential care and day activity centres compounds information loss. | Development: Piloted feedback from individuals, carers and ANDS MDT staff. Medical information removed at this stage as not updated.<br><br>Wider Intervention Name: N/A<br><br>HP: Communication Passport.<br><br>Other Element: N/A<br><br>Training: "Education and training on the use of the communication passport was given to caregivers and/or family members during the inpatient stay and at the time of discharge." (p.167–168)<br><br>Intended Use: Community Caregivers. | Explicit:<br><br>Passport contents individualised following observation and discussion with staff and carers<br><br>HP to improve communication (infer by increasing knowledge) between Autistic/ID people, community care and health providers to address issues arising from challenging behaviours and communication difficulties.<br><br>Implicit:<br><br>Accessibility to patients and carers prioritised (e.g: using<br><br>coloured pictures, simple language)<br><br>Written in first person language and holistic (ie: including thing that make the person happy) to humanize.<br><br>HP updated during inpatient stay to increase relevance. | Quantitative: N/A<br><br>Qualitative:<br><br>Some caregivers and institutions were unequipped to use the recommended communication strategies in the HP.<br><br>Informal feedback (from caregivers, family and healthcare professionals) indicates the HP is significantly helpful in aiding understanding and patient care, including information on: sensory, communication profile, functional level and specific behavioural triggers.<br><br>HP is useful for ANDS ward staff, as well as its intention for community carers. |

*(Continued)*

**Table 4.** (Continued)

| Author: | Context: | Intervention: | Mechanisms: | Outcomes: |
|---|---|---|---|---|
| Unitt [57] | Population: LDs. | Development: N/A | Explicit: | Quantitative: N/A |
| | Setting: Hospitals. | Wider Intervention Name: N/A | HP akin to "instruction manual" (p.1); enhances knowledge of patients' needs—important for those who lack capacity. | Qualitative: N/A |
| | Staff: Unclear. | HP: Hospital Passports completed by patient/carer. | | |
| | Known Issues: Hospitals as an alien, terrifying environment. | Other Element: Acute liaison nurses sometimes available to aid care. | Increase safe, person-centred care and improve patient experience if used correctly, including physical placement | |
| | Patients as heterogeneous. | Legally entitled to reasonable adjustments. | Misuse is a safety issue. Including; HPs lost, filed without being read; not accessible; lack of staff time/inclination to read. | |
| | Acute LD liaison nurses not always available. | Training: N/A, but LD training now available on university health professional courses | | |
| | Mental Capacity Act: may not always have capacity to make own medical decisions. | Intended Use: Nurses, but relevant to "Healthcare assistants" and "Senior Consultants." (p.2) | Raising awareness of HPs is required. | |
| | | | Accountability will increase use, so not seen as an 'optional extra'. | |
| | | Written in conjunction with an individual, their families, carers and friends" | Implicit: N/A | |

Key: ASD: Autism Spectrum Disorder; ID: Intellectual Disability; LD: Learning Disability/Difficulty; CBPR: Community-Based Participatory Research.

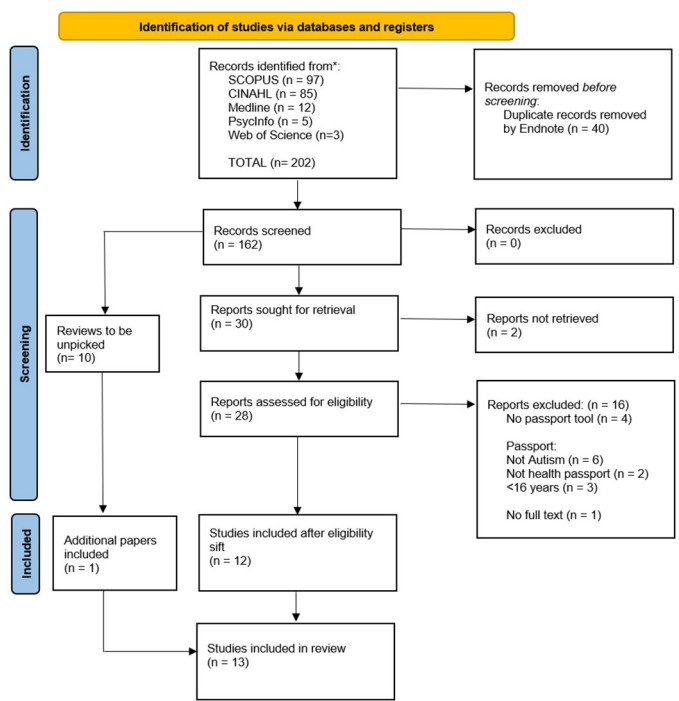

**Fig 1. PRISMA flow diagram study design, methodological quality, populations and settings.**

implied by the researchers than explicitly noted by authors, and the vast majority of these did not reference a particular theory. Only 4 sources included quantitative **O**utcome measures, and those which did tended to report low usage of HPs. Only two sources contained sufficient detail to allow the development of full CMO configurations; the remaining sources lacked **O**utcomes measures that were directly associated with **M**echanisms.

**Context.** In this section we discuss the **C**ontext in terms of population, setting, staff and known issues. Within the 13 sources, the most frequently discussed intervention target population were Autistic people (n = 8), with other populations including individuals with ID/IDD (n = 5), Parents or Families (n = 2), children (n = 1), those with severe disabilities (n = 1), professionals (n = 1) and those with LDs (n = 1) (see Table 4). Among papers that described research participants in the evaluation of HPs (n = 5), the demographics, and sometimes even the number, of participants in these samples, were often poorly described. Four papers which detailed participants, listed the professionals involved, [30, 47–49] with one mentioning a multidisciplinary team but not specifying any further details [51]. Two papers mentioned age, [30, 47] sex [30, 47] and ethnicity [30, 48] of participants. Nicolaidis et al. described their participants the most, including race, participant education level, living arrangement and level of assistance needed [30].

Most papers (n = 10) focused on a hospital setting, including hospitals in general (n = 4) and emergency departments (n = 2). Staff members anticipated to be recipients of HPs included hospital staff, community staff, or health professionals in general.

There was considerable overlap in the known contextual factors affecting healthcare equity for Autistic adult patients, including a lack of staff training (n = 11) and corresponding knowledge of frequent co-occurring conditions (n = 10). Specific barriers identified as impacting Autistic healthcare experiences included communication differences (n = 6) and sensory

difficulties (n = 2). It is important to note that these barriers are not caused by the Autistic qualities or Autistic people themselves, but by the interaction of these traits within poorly adapted services and environments [8]. Barriers within healthcare systems included staff shortages resulting in a lack of time (n = 5) and inaccessible or inappropriate hospital settings (n = 4). Contextual challenges were identified during the transitional period from paediatrics to adult care (n = 2) [47, 48]. Furthermore, patient dissatisfaction and a need for wider systematic change [52] were also described as known issues.

**Interventions.** When reviewing interventions, we included both HP tools, and wider interventions around HP tools, where applicable (see Table 4). The majority of sources (n = 7) did not describe the development of their intervention. Others provided limited details, such as naming the team the HP was developed by, [51] but not necessarily *how* this was achieved. Three papers mentioned an element of co-production, such as the use of service users in the evaluation, [30, 51, 53] or development of the tools [30, 53]. Two papers mentioned the use of a pilot as part of the HP development [30, 51]. Four sources noted that the HP was an aspect of a wider intervention, including projects focused on: transition from paediatrics to adult primary care services, [48] improving quality of care [34, 55] or as part of a healthcare toolkit [30].

There was variation in the depth of HP descriptions, with some sources containing detailed information, including visual representations of the tool itself, [48, 50] whereas others only mentioned a few elements within the HP [56]. Table 5 compares the HPs contents against the elements identified by Northway et al. [11] in their review of HPs for people with learning disabilities. Erickson Warfield et al. [47] and Perkins and Vanzant [52] were excluded from Table 5 as they both included descriptions of more than one tool. The most frequently included elements in HPs were levels of communication (n = 7), name (n = 5) and date of birth (n = 5). We interpreted "maintaining comfort" as attempts to reduce distress, for example: "ways to help me avoid distress", [50] and "accommodations to help patient stay calm and comfortable" [30]. Some HP elements described by Northway et al. [11] were not included in any of the tools, including: advanced care planning (do not resuscitate orders), next of kin, contacts in relation to discharge, requirements for an interpreter, how I communicate when I feel well or unwell, the person completing the form, oral hygiene needs, risk assessments, reasonable adjustments required, best interests meeting and/or decision, sleeping, behaviour or support with toileting. However, in addition to the elements listed by Northway et al., [11] the following items were identified from our included sources: impairments, triggers, interests, strengths, personality, physical placements for passport and ethnicity.

Three sources specified that HPs were intended to be completed by patients and caregivers [51–53] and three sources by staff [55, 51, 57]. In terms of training to facilitate use of HPs, six sources specified training, six did not and one was unclear as to whether it was provided. Training was provided for staff (n = 5), family and caregivers (n = 4) and Autistic individuals (n = 1), with some projects offering training to two or more stakeholder groups. Other papers focused on a wider intervention, with less of a descriptive focus on the HP tools themselves. For example, Harris et al. [48] noted the use of a HP to aid with transitions to adult care but did not include a reminder to complete a HP within their own transition checklist aimed at clinicians. Two sources did not describe the contents of the HPs discussed [34, 57].

**Mechanisms.** For two papers no information was provided as to the intended use of the intervention [48, 49]. Within our evaluation, we divided **M**echanisms into those stated explicitly by authors and those implied by the research team. 11 papers reported at least one explicit **M**echanism, however, **M**echanisms were rarely clearly stated, and often lacked detail. Explicit **M**echanisms that were based on robust theorisation were largely absent. An exception was Nicolaidis et al., [30] who utilised self-efficacy as part of their theory of change. Some additional explicit **M**echanisms described within this study were the use of personalised patient information as an

acknowledgement of heterogeneity within the Autistic population and identifying "Potential leverage points" (p.1181) from the research to explore in future practices. Also worthy of note, Lalive d'Epinay Raemy and Paignon [55] described multiple explicit **M**echanisms (n = 8), including the need for establishing trust and improved communication between disabled individuals and professionals, reducing emergency admissions, and improving the accessibility of hospital buildings. Additionally, Heifetz and Lunsky [49] included several explicitly stated **M**echanisms (n = 7), including the use of an implementation champion to aid embeddedness, and the suggestion that a co-ordinated approach and higher awareness of the tool would increase usage, improve embeddedness with staff and reduce the repetition of information by patients and carers [49]. An additional three papers described "some" theory behind the intervention.

Across the included sources, the most frequently described explicit **M**echanisms included: improving knowledge for both patients and clinicians (n = 4), strengthening care and person-centred practices (n = 4), and embedding the tool within systems (n = 3). Other potential **M**echanisms were poorly described and under theorised, for example, expecting that a HP would "increase quality of care", [50] "reduce anxiety for families" [54] or "improve communication" [55] without providing further details as to *how* this would be achieved. Our analysis of the sources resulted in a range of implicit **M**echanisms being generated by the research team including: increasing accessibility (n = 2), humanising the Autistic patient (n = 2) and ensuring the tool is embedded within the system (n = 1). Other implicit **M**echanisms mentioned include creating more holistic care [51], empowering staff and increasing quality of care, [53] changing staff behaviour [56] increasing acceptability through professional endorsements therefore increasing the acceptability of the tool, [52] easing transition between services [47] and updating the tool regularly to ensure relevance [51]. Implicit **M**echanisms were not derived from five papers (Table 4).

**Outcomes.** Of the 13 sources included, four contained at least one quantitative **O**utcome. Half of the papers did not include either quantitative or qualitative **O**utcome data (n = 7). Most quantitative **O**utcomes were focused on usage of HPs, the majority of which reported low usage. For example, Heifetz and Lunsky [49] demonstrated that from their small number of participants (n = 28), 82% did not have a chance to use their HP in the one-month follow-up period. Likewise, the transition checklist included in Harris et al. [48] intervention was shown to clinicians in 100% of appointments, aiming to prompt discussion about transition to adult healthcare services, however it did not include reference to their HP tool and was completed by clinicians in only 44% of appointments. In addition, Nicolaidis et al. [30] developed and tested an intervention which resulted in significant reductions in barriers to care, increased self-efficacy and higher a satisfaction of patient-practitioner communication. In this study, Autistic individuals said that the toolkit was useful (95%), and that they would recommend the toolkit to a friend (92%), or healthcare professional (95%) [30]. Two studies reported on usage without including numerical **O**utcomes. First, Lalive d'Epinay Raemy and Paignon's evaluation [55] of the Disability Admission sheet stated that it had been completed for nearly all disabled individuals within Geneva. Second, by contrast, within Kelbrick [34] usage of the HP tool was described as "limited".

Seven sources included qualitative data focused on the views and experiences of those using the HP or wider intervention. Within these studies, a range of methods and participant groups were used, with some qualitative **O**utcomes reported by authors without obvious sources for their assertions (see Table 2). Qualitative findings included HPs facilitating better communication, [49] and increased understanding of the patient [57]. However, there were concerns regarding the practicalities of implementation surrounding the lack of staff awareness of HPs [49] and the inexperience of the professionals who were using them [57]. Several barriers to HP usage were noted by families and healthcare professionals such as the additional time needed to implement accommodations, Nico 2016 and reduced physician engagement [48].

**CMO configurations.** CMO configurations are created to understand causality within realist evaluation and to identify if an intervention is working, in which **C**ontexts, for whom, and to what extent [41]. This is achieved through consideration of the underlying **M**echanisms that were or were not triggered in the evaluation **C**ontext [58]. Due to a lack of theories of change and/or **O**utcome measures, the majority (n = 11) of the literature reviewed was unsuitable for generating CMO configurations. In Box 1 we present CMO configurations for Nicolaidis et al. [30] and Harris et al. [48]. We do not feel that it would be appropriate to create an overarching CMO configuration for HPs, due to the weakness of the included evidence.

## Discussion

This realist review included 13 sources focused on HPs for Autistic adults. The papers were of varying quality with only one meeting all eight criteria we included in our quality assessment for realist review. The majority of sources described contextual information and at least some intervention details, but most were lacking in **M**echanisms. Only four sources included quantitative **O**utcomes. Nicolaidis et al., [30] was the most highly rated source in quality, and demonstrated increased measures of self-efficacy, reduced barriers to care and increased satisfaction following appointments amongst passport users. However, within this study, it is unclear as to how much of the impact was related to the HP tool, compared to the wider toolkit intervention.

The UK Medical Research Council guidance for the development and evaluation of complex interventions [59] provides a framework that researchers can follow to develop well-theorised interventions. This includes identifying problems, developing theoretically informed potential solutions, and evaluating those interventions, to ensure they have the best chance of succeeding in their current **C**ontext. Within our review, no paper other than Nicolaidis et al. [30] showed clear adherence to this iterative process. Other sources were identified as quality improvement initiatives, which busy health professionals juggled alongside their clinical duties, often relying on "common sense" and "ground-up" approaches due to high workloads impacting on time available to develop and evaluate interventions [60]. Furthermore, where interventions are moved to new **C**ontexts, it is acknowledged that they may need to be redesigned and subjected to further evaluation, [61] for example, Heifetz and Lunsky [49] found that when three groups of patients were asked to update an existing tool, it varied significantly across the three Canadian study sites.

Both government policies [24] and much of the literature we reviewed suggest that HPs improve care for Autistic people and should be widely utilised. Although HPs might have potential, their use and efficacy is still in its infancy and the varying quality of current research impedes the replication and evaluation of these studies. The literature included within this review described barriers to accessing care, including communication, staff attitudes and training. Our realist evaluation has therefore shown that the recommendation to roll out HP tools to reduce health inequality for Autistic adults is currently unwarranted. Despite the current lack of widespread evidence of efficacy, our analysis identified some potential in interventions which included a HP alongside a wider toolkit. For example, those which included service users in intervention development could be an important mechanism to drive change [30, 55]. Furthermore, recurrent contextual barriers to high-quality healthcare for Autistic adults, and **M**echanisms that were shared across multiple interventions show that there is some common ground on which tools can be developed.

However, any benefits have not been seen consistently in relation to asthma [17], and there is also conflicting evidence regarding the value and feasibility of birth plans [21]. A realist lens would suggest that even when there is a 'life or death' clinical implications of not using a tool

**Table 5. Elements included within HPs.**

| | Blair (2013) [53] | Brasher et al. (2020) [50] | Haidrani (2017) [54] | Harris et al. [48] | Heifetz and Lunsky (2018) [49] | Lalive d'Epinay Raemy and Paignon (2019) [55] | Nicolaidis et al, (2016) [30] | LDP (2014) [56] | Kelbrick et al, (2014) [34] | Sajith, Teo and Ling (2018) [51] | Unitt (2018) [57] | TOTAL per element |
|---|---|---|---|---|---|---|---|---|---|---|---|---|
| Level of Communication (Expression and Understanding) | ✓ | ✓ | | ✓ | ✓ | | ✓ | ✓ | | ✓ | | 7 |
| Date of birth | ✓ | ✓ | | ✓ | | | ✓ | | | ✓ | | 5 |
| Name | ✓ | ✓ | | ✓ | | | ✓ | | | ✓ | | 5 |
| Contact Person | ✓ | ✓ | ✓ | ✓ | | | ✓ | | | | | 5 |
| Communication of pain and distress | | ✓ | | ✓ | | | ✓ | ✓ | | | | 4 |
| Sensory Impairments | | ✓ | | ✓ | | | ✓ | | | ✓ | | 4 |
| Things I like/don't like | ✓ | ✓ | | | ✓ | | | | | ✓ | | 4 |
| Medical history/ health information | | ✓ | | ✓ | ✓ | | | | | ✓ | | 4 |
| Current medication | | ✓ | | ✓ | ✓ | | | | | ✓ | | 4 |
| Level of support required with nutrition | | | | ✓ | ✓ | | | | | ✓ | | 3 |
| Maintaining comfort needs | | ✓ | | | | | ✓ | | | | | 2 |
| Contact Details | ✓ | ✓ | | | | | | | | ✓ | | 3 |
| Mental Capacity Assessment | ✓ | ✓ | | | | | ✓ | | | | | 3 |
| Mobility | | | | ✓ | | | ✓ | | | | | 2 |
| Date of form completion | | | | ✓ | | | ✓ | | | | | 2 |
| Name of General Practitioner | | ✓ | | ✓ | | | | | | | | 2 |
| Personal Care | | | | ✓ | | | | | | | | 1 |
| Allergies | | | | ✓ | | | | | | | | 1 |
| Support with taking medication | | | | ✓ | | | | | | | | 1 |
| Religion | ✓ | | | | | | | | | | | 1 |
| National Health Service Number | | ✓ | | | | | | | | | | 1 |
| Other professionals involved | | ✓ | | | | | | | | | | 1 |
| GP Contact Details | | | | ✓ | | | | | | | | 1 |
| Power of Attorney | | | | | | | ✓ | | | | | 1 |
| Independent Mental Capacity Advocate | ✓ | | | | | | | | | | | 1 |
| **TOTAL per source** | 9 | 15 | 1 | 16 | 5 | 0 | 11 | 2 | 0 | 9 | 0 | |

*[52 and 47] excluded as they both included descriptions of more than one tool.

that a sufficiently challenging context can reduce the feasibility of using tools. This has been noted by Nicolaidis et al: *". . .It is likely that an accommodation report may not be (. . .) sufficient to eliminate all constraints affecting PCPs' ability to care for Autistic patients. . ."*[30, p.1188]. We summarised the contextual barriers we extracted and inferred from included sources graphically in Fig 2.

Implementing a HP tool into healthcare systems will have limited impact if the wider systems are not also changed as care is *". . .a complex interplay between an individual's Autistic characteristics, the healthcare provider's knowledge and attitudes about Autism, and the healthcare system"*[30, p.1181]. For HPs to achieve patient benefit, there is a need for systems that allow healthcare practitioners to engage with these tools in a useful and sensitive way, for example including longer consultation times and training to increase knowledge and confidence of how to support Autistic patients. If barriers to health professionals using HPs in consultations were removed, social factors such as the routine normalising of neurotypical behaviours and communication which result in the widespread stigmatisation of Autistic people would still be likely to influence the likelihood of Autistic individuals opting to self-disclose an Autism diagnosis. In a governmental review of 'Think Autism', [23] Autistic respondents felt that disclosure led to diagnostic overshadowing. Some Autistic people also have reported withholding their diagnosis in healthcare appointments as they felt it would negatively affect the treatment they received [62]. HPs cannot be embedded without a respect for Autistic communication. There is a need for a cultural change to reduce stigmatisation by healthcare staff [7] and provide a supportive sensory environment for Autistic people [63]. Accordingly, training and resources for clinicians must be developed and delivered by Autistic people, based on a neurodivergent-affirming model of Autism, as has been present in the Social Model of Disability since the 1960s. Working with Autistic clinicians will also increase the likelihood of acceptable interventions [64].

However, we acknowledge that this will not be easy to achieve; in one of the papers within this review a 5-day training course was initially envisaged but was replaced by 15 minute "on ward" sessions, [55] showing the significant pressures within the system and barriers to implementing training [65].

With regards to the hospital setting itself, there are significant sensory barriers for Autistic staff and patients [66]. Best practice guidance has been developed to help attend to sensory challenges within inpatient environments, [67] although this has not yet been routinely implemented. Part of making healthcare more accessible to Autistic people involves creating Autism friendly environments through cultural and systemic changes; for this to be more than tokenistic, Autistic voices must be centred within these efforts [68]. Organisations such as Autistic Doctors International [69] and the Maternity Autism Research Group [70] can play an important role in advancing change, due to members' dual status as clinicians and Autistic people.

## Clinical implications: Supporting Autistic patients

We conclude that HPs do not currently remove barriers to healthcare for Autistic people, and recommendations for their use are therefore inappropriate. At present suggestions that HPs can make healthcare more accessible and equitable for Autistic people are based on interventions that are largely atheoretical, small scale, and with poor embeddedness into healthcare practice. In addition, there have been no experimental studies of HPs. It has been suggested by Sharpe et al. [71] that healthcare for Autistic people could be usefully improved using a national primary care Autism register, removing the "hidden" element of Autism. This is similar to the National Patient Register employed in Sweden, [37] but in the Context of neuro-normativity and healthcare systems that largely do not meet the need of Autistic adults, we have serious

## Box 1: CMO configurations

**Nicolaidis et al.'s intervention** [30] was conducted within the United States, was intensively co-designed, using community-based participatory research (CBPR), and research evidence from the team's prior participatory studies. The low intensity online toolkit, which had separate versions for Autistic people and supporters, included a HP, health information for Autistic people and additional resources. The intervention was tested in a **Context** where Autistic patients had unmet and complex health needs, and primary care providers (PCPs) lacked knowledge, made assumptions, and were unwilling to make accommodations. **Mechanisms** that were triggered, resulting in desired **Outcomes,** included high levels of community acceptance of the toolkit, including ease of use and likelihood to recommend, which may have been achieved through the participatory design process. Additionally, the majority of participants agreed that their HP could be mailed to their PCP in advance of appointments, reducing barriers to use within consultations. Alongside acceptability to patients, the majority of PCPs found it useful as well. Accordingly, the toolkit increased Autistic patients' self-efficacy and reduced barriers to accessing healthcare. Among a minority of Autistic patients who saw their PCP within one month, satisfaction with care quality was increased. Qualitative data suggested that self-efficacy was improved by Autistic patients being aided by the tool to prepare for visits and allowing for more effective self-advocacy within appointments. Longer term follow-up of patients, and a more comprehensive evaluation from PCPs would have aided the strength of these findings.

**Harris et al.'s intervention**, [48] focused on the transition from paediatric to adult healthcare for Autistic young people and was developed by a "transition team" (p.755) that included neither young nor Autistic people but included the parents of an adult with cerebral palsy. The intervention was tested in a **Context** where adult PCPs were unwilling and untrained to care for Autistic adults, and Autistic young people had high rates of co-occurring conditions, including mental ill health. Furthermore, there was inadequate support for both groups around transition. **Mechanisms** were sometimes, but not always, triggered, impacting on **Outcomes**. For example, electronic prompts displaying transition checklists were displayed during every appointment, although lack of training for and/or confidence in PCPs, alongside inadequate appointment length, meant that this **Mechanism** did not result in the completion of the checklist, let alone the HP tool, more than half of the time. In recognition of the time constraints, PCPs could refer patients to a social worker for a specific transition appointment, which resulted in a social worker completing a transition checklist for almost half of eligible patients. However, the electronic prompts for PCPs (and presumably social workers) did not include reference to the HP element of the intervention, likely reducing the number of times it would have been discussed by PCPs and social workers. We therefore theorise this absence would reduce the number of times the HP tool would have been used by Autistic patients, although there is no **Outcome** data to support this. Additional potential **Mechanisms** around supporting transition did not have corresponding **Outcome** data including the impact of a half day training event for families.

ethical concerns with regards to identification, prejudice and stigmatising treatment which could arise from this. For example, research with new Autistic mothers has demonstrated that staff have threatened involuntary social care interventions which can have life-changing and serious implications for parents and baby [72]. Currently, most approaches with an aim to reduce health inequalities for Autistic people seek to improve current services and systems in a bolt-on fashion. For example, in addition to AHP tools, training of health professionals is routinely suggested as a relatively low-cost intervention that can improve healthcare [65]. For example, in addition to AHP tools, training of health professionals is routinely suggested as a relatively low-cost intervention that can improve healthcare [65]. However, if the quality of training surrounding this intervention is poor, it could reinforce existing negative stereotypes and not create any cultural change. Training which was, for example, designed and delivered by Autistic people, through a neurodiversity-affirming lens, could reduce inequalities.

For instance, the Oliver McGowan mandatory training in Autism and Learning Disabilities is being rolled out across the UK, but the evaluation of these training packages was limited. Evaluation was impacted because not all staff received training, response rates were unable to be calculated, small numbers of people completed the whole training package and therefore there was a low-quality evidence base from which to draw recommendations [73]. By contrast, the National Autism Trainer Programme is co-designed, co-produced and co-delivered by Autistic people, thus having greater potential to shift health professional views of Autistic patients [74].

Regardless of the success of training, however, we believe that the embeddedness of neuro-normativity throughout all aspects of the healthcare journey, from booking an appointment to following post-appointment instructions, and stigma towards Autistic people within healthcare settings mean that new approaches are needed. There have been examples of good practice reported in terms of developing primary care health services specifically to meet the needs of neurodivergent people, in a ground-up manner. For example, the All Brains Belong non-profit organisation propose a model of health which promotes inclusivity and does not include a default neurotype [75]. We believe that new models of service provision, developed with a sound theoretical basis and in consultation with Autistic people are an area of promise to reduce health inequalities.

## Limitations

Our review was based on 13 sources, the majority of which were of poor quality for realist evaluation. Only seven [30, 34, 47–49, 51, 55] could be considered research studies, and many of these contained limited methodological detail. This included more than half not describing their HP tool in detail. Furthermore, only two [30, 48] contained a process evaluation, showing that little is known about how these tools and wider interventions were embedded into local contexts. Despite this, the review process was high quality. First, a systematic literature review was undertaken, including independent review by two researchers. This independent review style was used throughout the realist evaluation. Realist evaluation was facilitated through the use of existing tools and the RAMESES II checklist [44]. Weaknesses included our use of evidence identified through systematic searching and 'sister' grey literature only. This means that our review identified many examples of ineffective practice and unanswered questions as to if HPs can have any benefit for Autistic patients. To answer some of these questions, we developed a questionnaire for Autistic adults about HPs, which will be reported separately. Additionally, we limited our database searches to the English language and papers published in the last 12 years. We acknowledge this may have caused some selective bias.

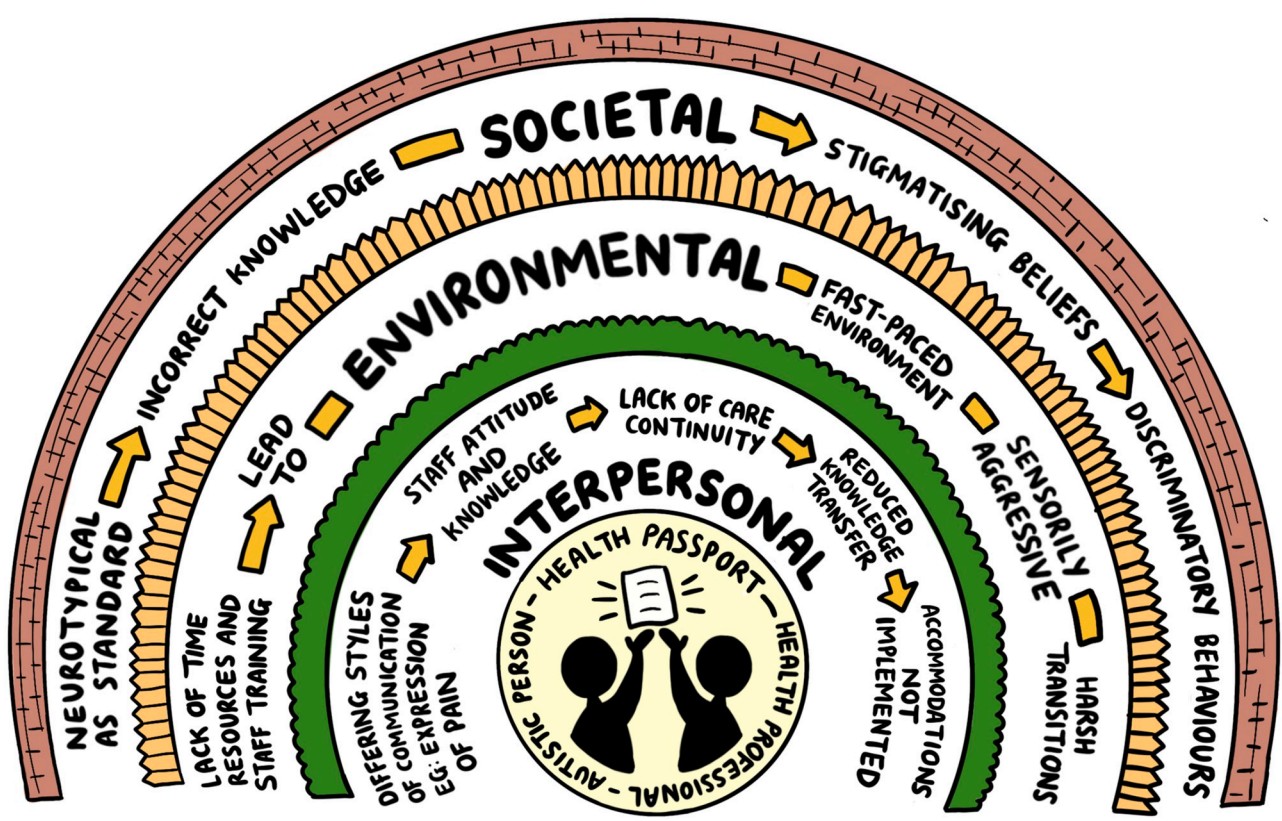

**Fig 2. Graphical representation to contextual barriers to AHP successful implementation.**

### Directions for future research

From the strength of the current evidence base and the significant barriers to HP use by clinicians and Autistic adults, we recommend that alternative interventions are sought to reduce health inequality for Autistic people. The barriers to HP use that we display in Fig 2 show that there are significant hurdles to overcome before a tool like a HP can flourish to the extent of, for example, an Asthma Action Plan. Accordingly, we recommend that future interventions should take account of the intervention **C**ontext to a much greater degree and must also be much more clearly based on theories of change, which are mapped to observable **O**utcome measures. Process evaluations should be embedded into designs to understand which elements of interventions are working well, which cannot be implemented in practice, and which are unable to overcome contextual barriers. Additionally unanticipated outcomes should be considered in evaluations. Furthermore, to ensure adequate understanding of who received interventions and who (if anyone) benefited from them, participant and clinician demographic should be collected and published. Many of the included sources contained qualitative findings, which can provide helpful contextual information and information on the feasibility and acceptability of implementing the intervention in the given **C**ontext; we recommend involving patients and clinicians in such evaluations, although it may be challenging to secure sufficient clinician engagement in evaluations of even well-designed studies [30].

The approach to future research that we have outlined fits well within Medical Research Council and the National Institute for Health Research guidance for intervention development

[59]. This evidence-based approach to developing interventions is significantly more costly than relying on busy health professionals who want to improve practice to develop interventions with very little resource. In America over 98% of funding for Autism research is directed to Autistic children, with very little investment in research to understand Autistic adults' lives and how to improve our health, despite a significant mortality gap [76]. Furthermore, in this landscape of underfunded research, lay Autistic adults are all too often excluded from research due to stigmatising attitudes that fail to see the value of Autistic people to research teams, and lack of widespread participatory research paradigms in clinical research [76]. There is some promise from co-produced approaches, however, we would argue the co-production process likely contains important mechanisms. HPs may not achieve the same outcomes independent of co-productive practices [77]. We recommend that details of co-productive practices should be clearly outlined in study outputs.

## Conclusion

Our review highlights that there is currently insufficient evidence, and the evidence that exists is low quality. Accordingly, we conclude that HP tools do not improve the accessibility of healthcare for Autistic adults. There is a legal duty in the UK, USA, and many other countries for services to make reasonable adjustments in order to ensure disabled individuals can access the healthcare they need, and it is clear that interventions to facilitate this for Autistic adults are very much in their infancy. To date, there has been inadequate inclusion of Autistic researchers and lay Autistic co-researchers in many studies; this is based on an outdated deficit-based understanding of Autism. If researchers are serious about improving Autistic adults' health, they must utilise strengths-based understandings of Autism which challenge neurotypical conventions and value the assets that Autistic people bring. When designing new tools to reduce the health inequalities Autistic adults face, researchers must take account of the barriers inherent in the intervention Context, ensure that appropriate theories are used in designing interventions, and that there is a clear map that shows the intended Mechanism of action. We are aware that Autism research currently tends to attract people from more privileged backgrounds. Reasonable adjustments are needed to ensure research conducted is accessible for Autistic individuals who wish to be involved in health intervention evaluation and development [78]. Interventions should be robustly evaluated in a way designed to assess *how* the intervention works, including showing if intended Mechanisms were triggered, and if any unintended consequences occurred, as well as measuring Outcomes related to health inequality for Autistic patients and knowledge and confidence for clinicians.

## Supporting information

**S1 Checklist. PRISMA 2020 checklist.**
(DOCX)

## Acknowledgments

We gratefully acknowledge support in developing our search strategy from Stephen Storey, a specialist librarian at Swansea University.

## Author Contributions

**Conceptualization:** Kathryn Williams, Amy Brown, Eleanor Healer, Aimee Grant.

**Data curation:** Rebecca Ellis, Kathryn Williams, Aimee Grant.

**Formal analysis:** Rebecca Ellis, Aimee Grant.

**Funding acquisition:** Kathryn Williams, Amy Brown, Eleanor Healer, Aimee Grant.

**Investigation:** Kathryn Williams, Eleanor Healer, Aimee Grant.

**Methodology:** Kathryn Williams, Amy Brown, Aimee Grant.

**Resources:** Amy Brown.

**Supervision:** Aimee Grant.

**Validation:** Rebecca Ellis.

**Writing – original draft:** Rebecca Ellis, Aimee Grant.

**Writing – review & editing:** Rebecca Ellis, Kathryn Williams, Amy Brown, Eleanor Healer, Aimee Grant.

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
