## [Decision Letter · Decision Letter 0]

13 Jun 2023

PONE-D-22-32749A hostile context, very limited intervention theory and almost no change in outcomes: findings from a realist review of health passports for Autistic adultsPLOS ONE

Dear Dr. Grant,

Thank you for submitting your manuscript to PLOS ONE. After careful consideration, we feel that it has merit but does not fully meet PLOS ONE’s publication criteria as it currently stands. Therefore, we invite you to submit a revised version of the manuscript that addresses the points raised during the review process.

We look forward to receiving your revised manuscript.

Kind regards,

Ali B. Mahmoud, Ph.D.

Academic Editor

PLOS ONE

2. Thank you for stating the following in the Acknowledgments/ funding Section of your manuscript:

“This project received funding from Swansea University’s Accelerate Health Technology Centre, which supported RE’s time on this project.”

“AG, KW, AB and EH received funding for this research from the Swansea University Accelerate Health Tech Centre (https://www.swansea.ac.uk/medicine/enterprise-and-innovation/business-support-projects/accelerate-healthcare-technology-centre/ ) Reference: 07/09/21. RE's time was funded by this grant. The funders had no role in study design, data collection and analysis, decision to publish, or preparation of the manuscript.”

3. We note that you have referenced (ie. Bewick et al. [5]) which has currently not yet been accepted for publication. Please remove this from your References and amend this to state in the body of your manuscript: (ie “Bewick et al. [Unpublished]”) as detailed online in our guide for authors

http://journals.plos.org/plosone/s/submission-guidelines#loc-reference-style "

Reviewers' comments:

Reviewer's Responses to Questions

**Comments to the Author**

1. Is the manuscript technically sound, and do the data support the conclusions?

Reviewer #1: Yes

Reviewer #2: Yes

2. Has the statistical analysis been performed appropriately and rigorously? 

Reviewer #1: N/A

Reviewer #2: Yes

3. Have the authors made all data underlying the findings in their manuscript fully available?

Reviewer #1: Yes

Reviewer #2: Yes

4. Is the manuscript presented in an intelligible fashion and written in standard English?

Reviewer #1: Yes

Reviewer #2: Yes

5. Review Comments to the Author

Reviewer #1: General comments: the review is clearly written, comprehensive and addresses a knowledge gap around an area of implementation that is being practiced in some settings despite a limited evidence base. The decision to use realist methodology extends this review beyond simply assessing the quality or extent of the evidence so that it is attentive to wider contexts and identifies the variety of mechanisms involved in the design and evaluation of health passports. The manuscript contains a strong level of detail on the characteristics of the included studies which support the authors’ observations about the overall lack of quality of the evidence base and limitations for drawing conclusions regarding the contexts, mechanisms and outcomes active in these approaches. This has helped to highlight how overall there has been a lack of attention to established theory in the development of these approaches and also importantly demonstrates mechanisms that may be overlooked, such as health professionals’ role in the process (e.g. their awareness of the tools and motivation/confidence to use them). The recommendations for future research are appropriate and consistent with the review findings but clinical implications could be expanded upon (see comments below).

Specific feedback:

Title: Could be more concise, or restructured to draw the immediate focus to the main purpose of the research (e.g. open with “A realist review of health passports for autistic adults” rather than the summary of the findings)

Background:

As the article and abstract open with “Autism is a normal part of cognitive diversity”, it may be helpful to use the Background to signpost towards literature on current understandings of neurodiversity for readers who may be less familiar and want to know more about this, as many readers from a medical background may be more familiar with the pathology paradigm.

“The case for a realist evaluation”: It may be useful here to reference realist theory and set the scene on what a realist review entails; the Context, Mechanisms and Outcomes framework is introduced without much context.

Methodology:

Exclusion criteria: The rationale for focusing only on adults and for omitting studies published before 2010 was not immediately apparent.

Table 2: Where the population in an included study is expressed as “people with intellectual disability” or similar, was the study included despite differences between autism and ID because the authors of original articles expressly included autism under their definition of ID or is this an assumption made by the reviewers? It seems there is some inconsistency about which populations are included in the review; perhaps it would be helpful to indicate more clearly whether the target population were sometimes included in broader samples.

Results

Context: The sentence “Within realist evaluation, Context is defined as the observable social, economic, political, and cultural structures which in turn inform if Mechanisms are triggered or not.” might be worth introducing earlier in the background or methodology section to help define the concepts involved in a realist review.

Where autism-related barriers are mentioned (e.g. communication, sensory differences), it may be worth emphasising that barriers for autistic people are not typically caused by autism itself but by autistic traits interacting with poorly-adapted features of services such as environments.

Figure 2: The findings in the ‘Context’ section reference Figure 2 but do not translate directly onto the concepts in the figure, which introduces some additional information beyond what is discussed in the results that appear to be drawn from the wider context of what is known from literature in this area - some of this is expanded upon in the discussion so may be helpful for the discussion to refer to Fig.2 where relevant, and for Fig.2 to be accompanied by an explainer signposting to relevant sections of the manuscript. The resolution of Fig.2 also makes it hard to read (including in the downloadable version).

Discussion:

The discussion of other tools that have been developed for different populations could be moved to the background section as it provides some relevant background but seems less relevant to the discussion.

“Accordingly, training…” – would it be possible to present some examples of how a neurodivergent-affirming/social model approach would look different and how this could be attended to in design of interventions and research.

The review highlights how coproduced approaches and attention to theories of change appeared to lead to more positive outcomes, but state overall that AHPs are ‘unwarranted’ and ‘do not improve the accessibility of healthcare’; would it be more appropriate to suggest that future research and decision-making requires refinement as well as more attention to the CMOs identified here? This could encourage future researchers and clinical decision-makers not to abandon these tools but to improve their usage and research, which would be more consistent with the recommendations made by the researchers in the section on future directions for research.

The clinical implications section appears less developed than other sections. Are there any tangible actions the authors can suggest to improve implementation of healthcare for autistic people based on the review findings? For example, Nicolaidis’s study suggested that the CBPR approach can potentially have some positive impacts when there is a clear theory of change, understanding of context and includes other supporting elements, and may benefit from further exploration. Are there any other approaches to addressing healthcare access and disparities for autistic adults that may work better or which still need evaluating, or are AHPs the best evidence we have at this time? What can clinicians/healthcare managers learn from the findings of this study that could improve the application of AHPs or similar tools they may be using in their practice (e.g. professionals need to have an awareness and understanding of their purpose and motivation/confidence to use them)?

Conclusions: “Researchers must take account of the barriers inherent in the intervention Context” – it may be useful to point out this can include barriers in the research design which may marginalise some autistic people from being involved in research into new interventions. Efforts should be made to reduce these as far as possible so samples are representative of the people for whom the interventions are designed and tested which will improve the applicability of findings to real life implementation.

Reviewer #2: Thank you for conducting this well-designed and clearly-reported systematic review. The only missing parts in your design were predetermined study types. Limiting included studies to the English language and to pre-2010 publications might have caused some selection bias. However, this does not affect the importance of this review.

You state that “This study was a systematic review that did not produce any new data”. Similar reviews were mistakenly called “empty reviews” in the past. I suggest changing this conclusion. Your review is pointing out an important area that has not been addressed in well-designed studies; preferably randomised controlled studies. You are making an excellent contribution through bringing this fact to the attention of researched, policy makers and funding bodies.

6. PLOS authors have the option to publish the peer review history of their article (what does this mean?). If published, this will include your full peer review and any attached files.

Reviewer #1: No

Reviewer #2: **Yes: **Adib Essali

---

## [Author Response · Author response to Decision Letter 0]

11 Jul 2023

We have attached a full rebuttal to the peer review comments which is attached as a "response to reviewers".

---

## [Decision Letter · Decision Letter 1]

11 Aug 2023

PONE-D-22-32749R1A realist review of health passports for Autistic adultsPLOS ONE

Dear Dr. Grant,

Thank you for submitting your manuscript to PLOS ONE. After careful consideration, we feel that it has merit but does not fully meet PLOS ONE’s publication criteria as it currently stands. Therefore, we invite you to submit a revised version of the manuscript that addresses the points raised during the review process.

We look forward to receiving your revised manuscript.

Kind regards,

Ali B. Mahmoud, Ph.D.

Academic Editor

PLOS ONE

Journal Requirements:

Reviewers' comments:

Reviewer's Responses to Questions

**Comments to the Author**

1. If the authors have adequately addressed your comments raised in a previous round of review and you feel that this manuscript is now acceptable for publication, you may indicate that here to bypass the “Comments to the Author” section, enter your conflict of interest statement in the “Confidential to Editor” section, and submit your "Accept" recommendation.

Reviewer #1: (No Response)

Reviewer #2: All comments have been addressed

2. Is the manuscript technically sound, and do the data support the conclusions?

Reviewer #1: Yes

Reviewer #2: (No Response)

3. Has the statistical analysis been performed appropriately and rigorously? 

Reviewer #1: N/A

Reviewer #2: (No Response)

4. Have the authors made all data underlying the findings in their manuscript fully available?

Reviewer #1: Yes

Reviewer #2: (No Response)

5. Is the manuscript presented in an intelligible fashion and written in standard English?

Reviewer #1: Yes

Reviewer #2: (No Response)

6. Review Comments to the Author

Reviewer #1: I am satisfied the authors have addressed all the reviewers’ comments. However, one point that I noted to be inconsistent with their other arguments is on p.45 line 595: “However, the quality of this, for example if it is designed and delivered by Autistic people through a neurodiversity affirming lens, can mean that such training could reinforce existing negative stereotypes and not impact on culture change” – did the authors mean to say that if training is not delivered in this way it may reinforce negative stereotypes? Please check this wording and adjust or clarify if necessary.

Reviewer #2: (No Response)

7. PLOS authors have the option to publish the peer review history of their article (what does this mean?). If published, this will include your full peer review and any attached files.

Reviewer #1: **Yes: **Charlotte Featherstone

Reviewer #2: **Yes: **Adib Essali

---

## [Author Response · Author response to Decision Letter 1]

19 Aug 2023

Please see attached word document.

---

## [Decision Letter · Decision Letter 2]

29 Aug 2023

A realist review of health passports for Autistic adults

PONE-D-22-32749R2

Dear Dr. Grant,

We’re pleased to inform you that your manuscript has been judged scientifically suitable for publication and will be formally accepted for publication once it meets all outstanding technical requirements.

Kind regards,

Ali B. Mahmoud, Ph.D.

Academic Editor

PLOS ONE

Additional Editor Comments (optional):

Reviewers' comments:

Reviewer's Responses to Questions

**Comments to the Author**

1. If the authors have adequately addressed your comments raised in a previous round of review and you feel that this manuscript is now acceptable for publication, you may indicate that here to bypass the “Comments to the Author” section, enter your conflict of interest statement in the “Confidential to Editor” section, and submit your "Accept" recommendation.

Reviewer #1: All comments have been addressed

2. Is the manuscript technically sound, and do the data support the conclusions?

Reviewer #1: Yes

3. Has the statistical analysis been performed appropriately and rigorously? 

Reviewer #1: N/A

4. Have the authors made all data underlying the findings in their manuscript fully available?

Reviewer #1: Yes

5. Is the manuscript presented in an intelligible fashion and written in standard English?

Reviewer #1: Yes

6. Review Comments to the Author

Reviewer #1: (No Response)

7. PLOS authors have the option to publish the peer review history of their article (what does this mean?). If published, this will include your full peer review and any attached files.

Reviewer #1: **Yes: **Charlotte Featherstone

---

## [Editor Report · Acceptance letter]

31 Aug 2023

PONE-D-22-32749R2 

A realist review of health passports for Autistic adults 

Dear Dr. Grant:

I'm pleased to inform you that your manuscript has been deemed suitable for publication in PLOS ONE. Congratulations! Your manuscript is now with our production department. 

Kind regards, 

on behalf of

Dr. Ali B. Mahmoud 

Academic Editor

PLOS ONE